

# Response of the subalpine bunchgrasses to wildfires and its effects in the relative abundance of the volcano rabbit in the Ajusco-Chichinautzin Mountain Range

Juan M. Uriostegui-Velarde[1], Alberto González-Romero[2], Areli Rizo-Aguilar[3], Dennia Brito-González[1] and José Antonio Guerrero[1]

[1] Facultad de Ciencias Biológicas, Universidad Autónoma del Estado de México, Cuernavaca, Morelos, Mexico
[2] Red de Biología y Conservación de Vertebrados, Instituto de Ecología A.C., Xalapa, Veracruz, México
[3] Facultad de Ciencias Químicas e Ingeniería, Universidad Autónoma del Estado de Morelos, Cuernavaca, Morelos, Mexico

Corresponding author
José Antonio Guerrero,
aguerrero@uaem.mx

## ABSTRACT

The volcano rabbit (*Romerolagus diazi*) is a lagomorph endemic to the central mountains of the Trans-Mexican Volcanic Belt and is classified as threatened at extinction risk. It is a habitat specialist in bunchgrass communities. The annual wildfires that occur throughout its distribution range are a vulnerability factor for the species. However, the effects of wildfires on volcano rabbit populations are not fully understood. We evaluated the occupancy and change in the volcano rabbit relative abundance index in the burned bunchgrass communities of the Ajusco-Chichinautzin Mountain Range during an annual cycle of wildfire events. Additionally, we assessed the factors that favor and limit occupation and reoccupation by the volcano rabbit using the relative abundance index in burned plots as an indicator of these processes. The explanatory factors for the response of the volcano rabbit were its presence in the nearby unburned bunchgrasses, the height of three species of bunchgrass communities, the proportion of different types of vegetation cover within a 500 m radius around the burned plots, heterogeneity of the vegetation cover, and the extent of the wildfire. Statistical analyses indicated possible reoccupation in less than a year in burned bunchgrass communities adjacent to unburned bunchgrass communities with volcano rabbits. The relative abundance index of volcano rabbits was not favored when the maximum height of the *Muhlenbergia macroura* bunchgrass community was less than 0.77 m. When the vegetation around the burned plots was dominated by forest (cover >30% of the buffer) and the fire was extensive, the number of latrines decreased per month but increased when the bunchgrass and shrub cover was greater around the burned plots. While the statistical results are not conclusive, our findings indicate a direction for future projects, considering extensive monitoring to obtain a greater number of samples that contribute to consolidating the models presented.

## INTRODUCTION

Wildfires produce variations in the space-time conditions of terrestrial ecosystems (*Díaz-Delgado et al., 2002*; *Banks et al., 2011*). In the last century, activities related to agriculture and livestock have altered the frequency, extent, and intensity of wildfires (*Bowman et al., 2009*). Moreover, climate change models predict wildfires of greater magnitude in certain areas of the planet due to increasing temperatures (*Cary & Banks, 2000*; *Bradstock, 2008*). The modification of the natural wildfire regime may cause ecosystems to lose resilience, resulting in the loss of biodiversity and affecting key ecosystem processes (*Menges & Hawkes, 1998*; *Díaz-Delgado et al., 2002*; *Rockweit, Franklin & Carlson, 2017*).

The effects following wildfire can be positive or negative, depending on the species present in an area and their surroundings (*Love & Cane, 2016*). Some ecosystems can tolerate wildfires. However, their recovery capacity depends on the frequency, extent, and intensity of the fire; for example, pine forests and alpine bunchgrass (*Rodríguez-Trejo & Fulé, 2003*; *Horn & Kappelle, 2009*). At the organism level, certain plants benefit from post-fire conditions, such as nutrient concentrations in the soil, increased light availability, and improved seed germination (*Chambers et al., 2007*; *Love & Cane, 2016*). On the other hand, for animals, the consequences are fatal for individuals who cannot escape wildfires (*Bunting, 1987*; *Whelan, 1995*), and some animals may modify their diet and behavior due to alterations in the plant community that result from wildfires (*Rhodes et al., 2010*).

Wildfires must be considered in habitat management strategies, so preventing wildfires may also have negative effects on certain ecosystems, resulting from the excessive accumulation of organic matter acting as fuel, leading to more intense fires that cover greater areas (*Grant et al., 2010*). The fact that an ecosystem can burn frequently also leads to negative effects on the structure and composition of the vegetation because it interrupts the vegetation successional cycle, reduces plant diversity, and results in a homogeneous vegetation association (*Jennings et al., 2016*).

The responses of plant communities to post-wildfire conditions have been studied extensively (*Whelan, Rodgerson & Dickman, 2002*; *Driscoll et al., 2010*; *Chia et al., 2016*), but few studies have reported the effect of fire on fauna (*Clarke, 2008*; *Fontaine & Kennedy, 2012*; *Van Mantgem, Keeley & Witter, 2015*). Studies have reported that wildfires modify the distribution and abundance of small mammals, and the probability of survival of small mammals increases if they can find refuge (*Banks et al., 2011*; *Griffiths & Brook, 2014*; *Knight & Holt, 2005*). Additionally, even though the response is variable in the temperate forests of North America, small mammals are the first to arrive, followed by the snowshoe hare (*Lepus americanus*), which is distributed in coniferous and boreal forests, after the vegetation has been regenerating for 11 to 25 years after the wildfire (*Paragi et al., 1997*; *Cheng, Hodges & Mills, 2015*). Moreover, in the boreal forests of North America, *Fisher & Wilkinson (2005)* analyzed how plant succession is related to the diversity of mammals, observing that the high degree of small mammal richness increases when the forest is mature 76 years after the wildfire.

After wildfires, habitat composition plays an important role in the survival of animals. In many cases, the surrounding environment that was not touched by the wildfire offers

shelter and is a determinant of the survival of individuals of different species (*Banks et al., 2011*). In the aftermath of wildfires, populations of small mammals are either extirpated locally or they recover, although the latter depends mainly on the presence of source populations (*Banks et al., 2011*). Additionally, the degree of recolonization depends on the regeneration of vegetation cover, which can boost recolonization (*Banks et al., 2011*). However, as the frequency of wildfires increases, populations of both mammals and vegetation progressively lose their ability to reestablish (*Díaz-Delgado, 2003*).

The volcano rabbit *(Romerolagus diazi)* is a lagomorph endemic to the central mountains of the Valley of México. This species is listed as Endangered by the IUCN (*Velázquez & Guerrero, 2019*) because it faces habitat loss caused by changes in land use (*Uriostegui-Velarde et al., 2018*). Volcano rabbit abundance is associated with the presence of bunchgrasses of the species *Muhlenbergia macroura*, *Festuca amplissima*, and *Stipa ichu* (*Velazquez & Heil, 1996*; *Hunter & Cresswell, 2015*; *Rizo-Aguilar et al., 2015*). The brushland and bunchgrass communities where they grow are affected by wildfires (Comisión Nacional Forestal; *García-Romero, 2004*), under the wildfire dynamic, these communities are prone to burning and become hazardous to the survival of individuals of different species and therefore its conservation (*Setterfield et al., 2010*; *Litt & Steidl, 2011*; *Balch et al., 2013*).

The volcano rabbit is a habitat specialist on bunchgrass communities and the annual wildfires that occur in its distribution range have become a factor of vulnerability for the species. In the memories of the International Workshop for the Conservation of Endangered Species of Mexican Lagomorphs held in 1996, it was shown that wildfires are the main cause of mortality for the volcano rabbit; consequently, an increase in wildfires could lead to local extinction of the species (*Portales et al., 1997*). However, the effects of wildfires on volcano rabbit populations are largely unknown. In contrast, one study conducted on the Iztaccíhuatl Volcano suggested that an increase in recent fire activity resulted in an increased abundance of rabbits because habitat quality improved due to reduced competition for bunchgrass (*Hunter & Cresswell, 2015*). Wildfire aftermath reports state that volcano rabbits returned to and recolonized the areas that they occupied before the fire (*Rangel-Cordero, 2008*; *Brito-González, 2017*). In this context, an analysis of the response of volcano rabbit populations to this annual phenomenon is of interest because it is an endangered species, and we need to understand how fires influence their distribution areas and abundance.

The objective of this study was to analyze the response of the volcano rabbit under post-wildfire conditions in areas of burned bunchgrass communities, and to compare the relative abundance index of this species between burned and unburned plots. Volcano rabbits form colonies and do not usually stray far from their burrows; therefore, the count of their latrines is an index of the relative abundance of the species (*Velázquez, 1994*; *Rizo-Aguilar et al., 2015*). It is expected that in the burned plots, the number of latrines will be close to zero; however, as bunchgrass coverage recovers, the number of latrines will increase. The increase in latrines will depend on the distance between burned and unburned plots. Bunchgrass growth has a positive relationship with the number of latrines. Finally, if volcano rabbits in the landscape could find shelter in unburned bunchgrass, as

the vegetation in burned areas regenerates, they would be able to return to the bunchgrass they inhabited before the wildfire. However, if volcano rabbits were not able to find refuge during the wildfire, they would probably not go back to the bunchgrass, even if the vegetation had regenerated.

## MATERIALS AND METHODS

### Study area

The study was carried out between March 2016 and March 2017 on the outskirts of the Pelado Volcano and in the surroundings of Las Palomas Hill, located in the Ajusco-Chichinautzin Mountain Range in the central mountains of the Trans-Mexican Volcanic Belt, in Mexico (Fig. 1). Part of the study area lies within two Protected Natural Areas: Lagunas de Zempoala and Chichinautzin Biological Corridor. Outside protected natural areas, there are human settlements where the main activities are growing crops and raising livestock. The dominant climate in the study area is Cb' (w2), semi-cold, subhumid, with a long cool summer, and a mean annual temperature of 5 °C to 12 °C; the temperature of the coldest month is between −3 °C and 18 °C, while the temperature of the warmest month does not exceed 22 °C, with rainfall less than 40 mm in the driest month, rainfall in summer, and a percentage of rain in winter that represents 5 to 10.2% of the annual total (García, 1988). It is located at an altitude between 2,975 and 3,155 m above sea level, is a rugged terrain where you can see several extinct volcanoes, the soil types are andosol and lithosol and the representative vegetation in the area is alpine and subalpine bunchgrass, conifers, mixed forest, live oak, cloud forest, scrub, aquatic and subaquatic vegetation, and crop fields (CONABIO, 2024).

### Sites selection

The locations and dates of the wildfires that occurred in the study area during 2016 were recorded by the Comisión de Recursos Naturales and the Comisión Nacional Forestal. A first visit was conducted to all areas where wildfires had been recorded during March and April 2016, and 10 sites were selected that had suffered wildfire incidents between March 26 and April 16, 2016, with the objective of obtaining homogeneous data, considering the date of the wildfire and the regeneration of the vegetation cover. The criteria for site selection were evidence of the presence of bunchgrass communities with traces of volcano rabbits (presence of fecal pellets or burned latrines), and a minimum distance between the sites of at least 1 km to maintain independence between them (Fig. 1). A distance of 1 km was considered because it is the minimum distance between the sampling units used in the monitoring programs of the species (Rizo-Aguilar et al., 2015; Uriostegui-Velarde et al., 2018).

At each of the 10 selected sites, two 25 × 25 m plots were set up: one in a burned bunchgrass community and the other in a nearby unburned bunchgrass (areas without burned plant matter that could act as a refuge) community that also had traces of volcano rabbits. On average, these two plots were 150 m (50–400 m) away from each other, and the unburned plot corresponds to the bunchgrass fragment closest to the burned plot with evidence of the presence of volcano rabbits, considering that the first rabbits occupying the
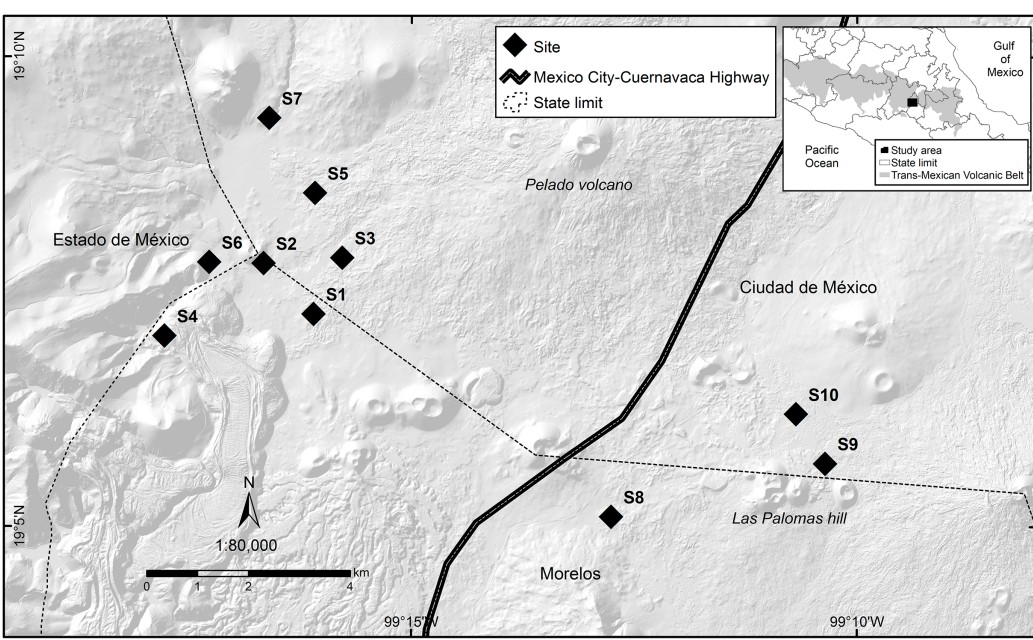

**Figure 1 Study area where the response of volcano rabbits to wildfires was assessed.** Numbers indicate the 10 study sites, each of which had two plots: one that had burned and one that had not.

burned plots will be those closest, located in unburned plots. Therefore, it is expected that the response of the burned plots is related to the relative abundance index (RAI) of the volcano rabbit in the unburned plots, while maintaining independence between sites with a possible interaction between plots. The first sampling was conducted in the second week of April. Plots were laid out with five stakes, one in the centre and the other four in the direction of the cardinal points (at approximately 17.67 m to delimit the 25 × 25 m plot) to mark the corners. The sites were visited every 30 days for 13 sampling visits.

## Relative abundance index of the volcano rabbit

The relative abundance index (RAI) of the volcano rabbit was estimated during each visit to the sites by counting the number of latrines in each plot. Volcano rabbits form colonies and they do not stray far from their burrows, so the presence of latrines indicates that individuals are distributed and frequently use those places (*Velázquez, 1994*; *Rizo-Aguilar et al., 2015*). The latrines were not removed from the plots so, to avoid counting the same latrine more than once, they were sprayed with different colors of odorless vinyl paint on each sampling visit. Since an indirect method was used, no animals were harmed in this study.

The differences in the RAI of the volcano rabbit were evaluated using a linear mixed-effects model (LME) implemented in the *nlme* package (*Pinheiro et al., 2017*) run in RStudio (Version 0.98.1028; *RStudio Team, 2014*). The number of latrines per plot (RAI) was the dependent variable and the condition, the distance between paired plots (burned and unburned), time and geographical position were the independent variables. We considered as fixed effects time nested in the condition burn and unburn of the plots to

analyze how the number of latrines changes between them plus distance between plots due to a possible effect of the distance between burned and unburned plots. To assess the relationship between plots, considering the scalar effect of the time and the geographical variation that cannot be controlled in the sites, and the possible effect masked by the distance between plots associated within each site and plots nested in site were treated as random effects. To normalize the residual value of the model, the number of latrines was log-transformed, adding 0.5 to all the data owing to the presence of zeros in the data. Finally, the most parsimonious LME was selected using the stepwise method.

## Relationship between bunchgrass height and relative abundance index of the volcano rabbit

Using a measuring tape, the height of five bunches of each of the bunchgrass species (*Muhlenbergia macroura*, *Stipa ichu*, and *Festuca amplissima*) was monitored every month during the study, and the height of the bunchgrasses was only averaged in the burned sites as a proxy for the recovery of the bunchgrasses. Subsequently, monthly sampling for the relationship between the average height of each bunchgrass species (independent variables) and the number of latrines (RAI) of the volcano rabbit (dependent variables) was analyzed with an LME using the *nlme* package in RStudio. Likewise, using the *stepwise* method, the most parsimonious LME was selected, considering the average height of each bunchgrass species found in a burned plot as a fixed factor, and the 10 sites were considered random effects. To adjust the residual value to the number of latrines of the volcano rabbit, 0.5 was added to the data, which was also log-transformed.

## Analysis of the landscape and the size of the wildfire during occupancy in the burned plots

Using ArcMap 10 software (*ESRI, 2012*) a 500 m buffer radius was set up around the burned plots. Considering that the area of activity of the volcano rabbit is 2,500 m$^2$ (*Cervantes & Martínez-Vázquez, 1996*), the buffer area is equivalent to 314 areas of activity depending on the landscape. For each buffer, land use and vegetation cover were determined, and the area (ha) of each category of use was estimated based on the classification of a Landsat 8 image (resolution 30 m) taken with the TIRS and OLI on February 19[th], 2015 (*United States Geological Survey (USGS), 2015*) and using ENVI v5.1 (*Uriostegui-Velarde et al., 2018*). Seven categories were considered for land use and vegetation: bunchgrass, bunchgrass with forest, forest with bunchgrass, forest, scrub, crops, and exposed soil (Fig. S1). The inverse of Simpson's index was used to evaluate the diversity in each of the buffer areas (*Nogués-Bravo & Pérez-Cabello, 2001*) and the distance between plots and the extent of the fire. Information on the size of the wildfires in the 10 plots was provided by the Comisión Nacional Forestal.

The evaluation of the effects of land use and vegetation cover, distance between plots, and size of wildfires was carried out in two steps. First, the number of variables was reduced using principal component analysis (PCA). Subsequently, the scores for the six first principal components as independent variables were analyzed using a generalized linear model (GLM) using the influence of the components on the monthly average of the

number of latrines (RAI) of the volcano rabbit in each of the burned plots as dependent variables. The monthly number of latrines was log transformed and 0.5 was added to each value to account for records with values of zero. The linear model was fitted using a GLM with a Gaussian distribution. This analysis was run using RStudio software (*RStudio Team, 2014*).

## RESULTS

### Relative abundance index of the volcano rabbit

In total, 2,585 latrines (258.5 +/−78.20 latrines per site in a year, considering the plots in burned and unburned conditions) were counted. At sites 4 and 9, only three and nine latrines were counted, respectively, while at site 3 there were a total of 771 latrines, of which 769 were in the unburned plot. Considering the total number of latrines during the sampling year per plot, an average of 159 (SE = 72.47) latrines were counted, ranging from 0 to 769 latrines over the year. In burned plots, the average was 99.5 (SE = 38.1) latrines, with a minimum value of 0 and a maximum of 332 latrines over the year (Table S1). The minimum and maximum values of the number of latrines indicated a heterogeneous response in the relative abundance index of the volcano rabbit.

The fixed effects of the LME indicated that the distance between plots did not affect the number of latrines over time; therefore, the variable was discarded from the final model. Concerning burned and unburned conditions, at the beginning of the sampling, a positive effect was noted in the unburned condition (LME; $t = 3.8209$; $P = 0.0002$) and a negative effect in the burned condition (LME; $t = -2.8577$; $P = 0.0189$). As time increased, in the unburned condition, the number of latrines remained the same (LME; $t = -0.6970$; $P = 0.5055$), and in the burned condition, the number of latrines increased (LME; $t = 2.5073$; $P = 0.0129$; Table 1), as seen in Fig. 2.

The structure of the random factors in the LME shows that the variance between sites and the variance of the time effect between sites have a correlation of 0.998, indicating that the time effect and the effect of the burned/unburned condition on the number of latrines (RAI) of the volcano rabbit are similar between sites. However, the correlation between the variance of the nested plots within the sites and the variance associated with the time effect was −0.441, so the response was heterogeneous in the plots nested in their respective sites over time, evidencing a random effect between plots nested in sites over time. Note that the RAI of the volcano rabbit in the burned plots for the eight locations had a statistically significant positive tendency. The only sites with a negative tendency in the burned plots were 9 and 3. In the unburned plots, the tendency was negative except for sites 1, 3, 7, and 8, where the slope was positive (Fig. S2).

The results of the model with estimated values and different magnitudes revealed three patterns in the response of the volcano rabbits (Fig. S2): Pattern 1. Over time, the RAI of the volcano rabbit increased in the burned plots, with a variable response in the unburned plots (sites 1, 5, 6, 8, and 10). Pattern 2. As time passed, the RAI of the volcano rabbit in the burned plots became steady but increased in the unburned plots (sites 3 and 7). Pattern 3. There was no time effect in either of the two plots (sites 2, 4, and 9).

**Table 1 Comparison of the relative abundance index of the volcano rabbit in burned and unburned plots.** The results of the Linear Mixed-Effects model are presented with the 10 sites and the twelve months of sampling as random effects. Statistically significant *p*-values are in bold.

| Effects | | | | | |
|---|---|---|---|---|---|
| Random effects: | | | | | |
| Formula: ~ Time \| Site | | | | | |
| | StdDev | Corr | | | |
| Intercept | 0.4007 | (Intr) | | | |
| Time | 0.0643 | 0.998 | | | |
| Formula: ~ Time\|Plot in Site | | | | | |
| | StdDev | Corr | | | |
| Intercept | 1.0971 | (Intr) | | | |
| Time | 0.1483 | −0.441 | | | |
| Residual | 0.8206 | | | | |
| Fixed effects: log (Latrines + 0.5) ~ Plot/Time | | | | | |
| | Value | SE | DF | *t*–value | *P*–value |
| Intercept | 1.5376 | 0.4024 | 218 | 3.8209 | **0.0002** |
| Condition burnt | −1.5436 | 0.5401 | 9 | −2.8577 | **0.0189** |
| Condition unburnt: Time | −0.0139 | 0.0555 | 218 | −0.2507 | 0.8023 |
| Condition burnt: Time | 0.1392 | 0.0555 | 218 | 2.5073 | **0.0129** |

## Relation between grass height and the relative abundance index of the volcano rabbit

The results of the LME on the fixed factors indicated that the intercept did not differ from 0. Regarding the effect of grass height, for all three species, the slope was significantly different from 0, with a positive effect between the number of latrines of the volcano rabbit and the height of *Muhlenbergia macroura* (LME; $t = 2.9939$; $P = 0.0034$) and *Stipa ichu* (LME; $t = 2.8952$; $P = 0.0046$), whereas *Festuca amplissima* had a negative effect (LME; $t = −2.7705$, $P = 0.0066$; Table 2). As the heights of *M. macroura* and *S. ichu* increased, the RAI of the volcano rabbit increased. In contrast, as the height of *F. amplissima* increased, the RAI of the volcano rabbits decreased (Fig. 3). The random effects indicated variance ($\sigma = 1.2496$) in the number of latrines between the burned plots in the locations. The residual variance for each location is 1.0784.

Notably, in the models, the estimated RAI of the volcano rabbit did not increase with the bunch height of *F. amplissima*. In the burned plots at sites 1, 5, 7, 8 and 10, there was a significant increase in the RAI of the volcano rabbit as *M. macroura* grew, and to a lesser degree, in the burned plot from site 6. The models indicate that when bunchgrasses reach a height of 0.77 m it is possible to find between five and 25 latrines. Although the effect of the height of the bunchgrasses is statistically positive for *S. ichu*, graphically, there seems to be a weaker relationship between the RAI of the volcano rabbit and the height of this bunchgrass compared with the effect of *M. macroura* (Fig. S3). That is, there were variation between sites and within sites.

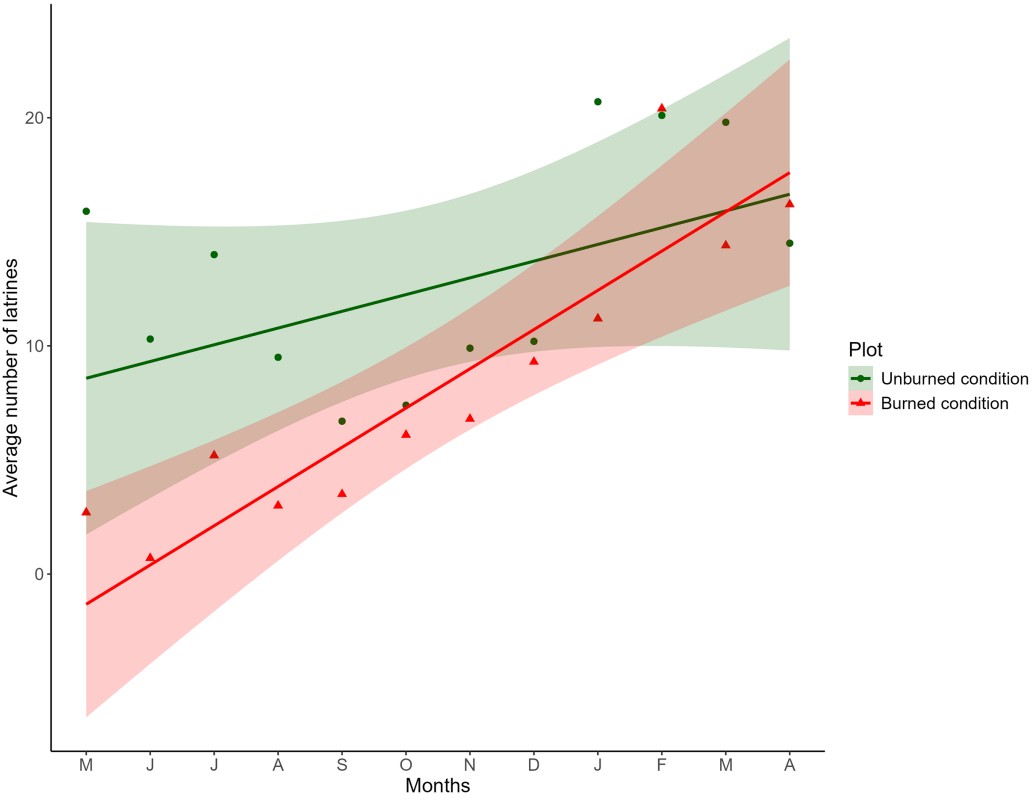

**Figure 2 Comparison of the monthly average of the number of latrines between burned and unburned plots in the Ajusco-Chichinautzin Mountain Range.** Sampling began in May 2016 and concluded in April 2017. The green circles represent the average number of latrines in the 10 unburned plots per month, and the red triangles represent the average number of latrines in the 10 plots burned per month. The lines show the estimated values based on the coefficients obtained from the mixed linear effects model, the shadow represents 95% confidence intervals.

**Table 2 Analysis of volcano rabbit relative abundance index concerning bunchgrass height in the Ajusco-Chichinautzin Mountain Range burned plots.** The bunchgrasses measured for comparison were *Muhlenbergia macroura* (mul), *Festuca amplissima* (fes) and *Stipa ichu* (sti). The sites were considered a random effect in the model. Statistically significant *p*-values are in bold.

**Effects**

Random effects:

Formula: ~ 1 | Site

|  | | Intercept | Residual |
|---|---|---|---|
| SD | | 1.2496 | 1.0784 |

Fixed effects: log (Latrines + 0.5) ~ mul + fes + sti

|  | Value | SE | DF | *t*–value | *P*–value |
|---|---|---|---|---|---|
| Intercept | −0.6609 | 0.6578 | 107 | −1.0047 | 0.3173 |
| mul | 0.0244 | 0.0081 | 107 | 2.9939 | **0.0034** |
| fes | −0.0153 | 0.0055 | 107 | −2.7705 | **0.0066** |
| sti | 0.0204 | 0.0070 | 107 | 2.8952 | **0.0046** |

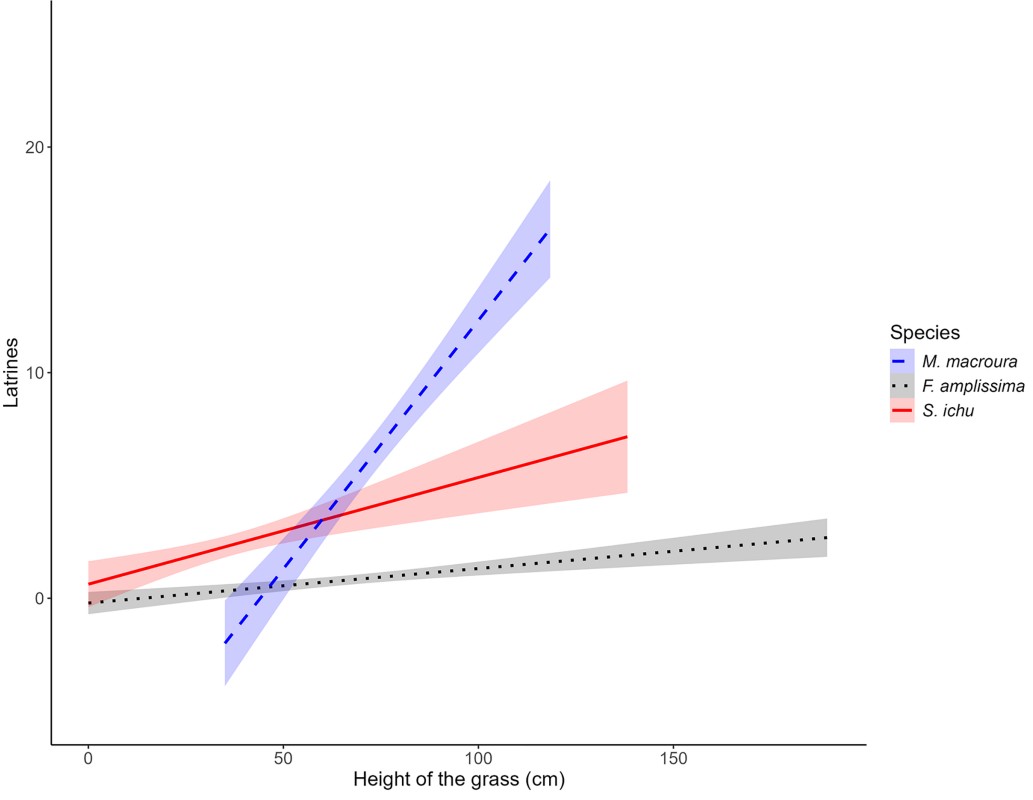

**Figure 3 Relationship between the average number of latrines of the volcano rabbit and the height of the bunchgrasses in the burnt plots in the Ajusco-Chichinautzin Mountain Range.** The graphs show the estimated values of the number of latrines in response to the height of the bunchgrasses based on the coefficients obtained from the Linear Mixed-Effects model, the shadow represents the 95% confidence intervals.

## Analysis of the landscape and wildfire size

On average, the burned plots had 0 to 27.7 latrines. The burned plots at sites 4, 3, and 9 generally had fewer latrines throughout the year (0, 0.17 and 0.75 latrines/month respectively). In contrast, the burned plots at sites 10, 8, and 1 had average of 27.7, 23.6 and 13.1 latrines/month, respectively. PCA indicated that six components explained 96.5% of the variation in the data (Table S2). PC 1 was negatively associated with crop cover and positively associated with the index of diversity of vegetation cover and the area of forest with bunchgrass. PC 2 was negatively associated with bunchgrass with forest and positively associated with exposed soil and scrub. PC 3 was negatively associated with the size of the wildfire and positively associated with the bunchgrass cover. PC 4 was negatively associated with the distance between plots and positively associated with bunchgrass. PC 5 was positively associated with the size of the wildfire and negatively associated with the forest cover. PC 6 was positively associated with cultivated areas and negatively associated with size of the wildfire.

In the GLM, the most parsimonious model ($AIC_c = 27.99$) included all six principal components with an explained deviance of 89.6%. The results (Table 3) indicated that the intercept was significantly different from 0 (GLM: $t = 5.528$; $P = 0.0117$). For the slope, PC3

**Table 3 Comparison of mean volcano rabbit relative abundance index with the estimated values for the principal components.** The principal component 3 (PC3) is the only one with a statistically significant effect on the relative abundance of the volcano rabbit. Statistically significant p-values are in bold.

|  | Estimate | SE | t–value | P–value |
|---|---|---|---|---|
| Intercept | 1.4064 | 0.2544 | 5.528 | **0.0117** |
| PC1 | 0.2678 | 0.2682 | 0.998 | 0.3917 |
| PC2 | −0.5745 | 0.2682 | −2.142 | 0.1216 |
| PC3 | 1.0286 | 0.2682 | 3.835 | **0.0312** |
| PC4 | −0.4523 | 0.2682 | −1.686 | 0.1903 |
| PC5 | −0.2549 | 0.2682 | −0.95 | 0.4121 |
| PC6 | 0.3519 | 0.2682 | 1.312 | 0.2808 |

are significant (GLM: $t = 3.835$; $P = 0.0312$). The monthly average number of latrines of the volcano rabbit is positively related to PC3, which is negatively associated with the size of the wildfire and positively associated with bunchgrass. Together, these components explained 18% of the variance, indicating a decrease per month in the number of latrines in burned plots as a function of the size of the wildfire; the number of latrines increased as bunchgrass on the landscape increased.

## DISCUSSION

Our results indicate that wildfires in the bunchgrasses of the Ajusco-Chichinautzin Mountain Range modify the distribution and abundance of the volcano rabbit. As volcano rabbits depend on bunchgrasses to build their burrows and feed on (*Cervantes & Martínez-Vázquez, 1996*; *Velázquez, Romero & León, 1996*; *Martínez-García et al., 2012*; *Rizo-Aguilar et al., 2015*), the lack of this resource can prompt their relocation to other areas where the bunchgrass cover was not reached by wildfires. Our data showed that the response of this lagomorph to wildfires is associated with at least three factors: the availability of bunchgrass areas to act as refuge during wildfires, the abundance of volcano rabbits in unburned areas, and the recovery of the height of the bunchgrass of *M. macroura*.

In bunchgrass areas near wildfires, unburned plots that were considered refuges for the volcano rabbit were limited. Hence, they are suitable shelter areas for animals capable of escaping wildfires. Shelters are fundamental to the recovery and reconstruction of ecosystems when a disturbance occurs (*Elmqvist et al., 2002*). Given the biological and behavioral characteristics of the volcano rabbit, our results provide evidence that the reoccupation of the five burned plots was an effect of the volcano rabbits inhabiting unburned plots. Consequently, the RAI of volcano rabbits increased in the burned plots, whereas in the unburned plots, abundance either decreased or remained constant. In other small mammal species, such as the agile antechinus (*Antechinus agilis*) and bush rats (*Rattus fuscipes)*, shelter availability and the survival of the remaining populations were found to be positively associated with the recolonization of burned areas after a wildfire (*Banks et al., 2011*). Our results contrast with those obtained by *Fisher & Wilkinson (2005)*,

who reported that in temperate forests, lagomorphs return to burned areas 11 to 25 years after the fires. The response of the volcano rabbit is like that of small mammals that reoccupy habitats in a heterogeneous manner immediately after fires (*Zwolak, 2009*).

No traces of the presence of volcano rabbits were found in three of the burned plots during the sampling year. This is probably associated with the low number or absence of latrines recorded in their corresponding unburned plots, suggesting that the possibility of reoccupation in wildfire areas depends on the abundance of volcano rabbits in the areas untouched by the wildfire. Another characteristic of the unoccupied burned plots was that the maximum height of *M. macroura* bunchgrasses was less than 0.77 m. The only exception was the burned plot in area 9, where its maximum height was 1.01 m, and no latrines were recorded.

In mammals, recolonization is limited when key attributes of the habitat are eliminated (*Lindenmayer et al., 1999*) and successful reoccupation relies on the recovery of vegetation cover (*Jennings et al., 2016*, *Fox, Taylor & Thompson, 2003*). For example, for the American hare, *Lepus americanus*, after a wildfire the response of populations has been associated with variations in undergrowth and canopy cover (*Mowat & Slough, 2003*; *Hodges, Mills & Murphy, 2009*; *Hodson, Fortin & Bélanger, 2011*; *Strong & Jung, 2012*; *Cheng, Hodges & Mills, 2015*).

Our analyses indicate that the RAI of volcano rabbits in burned plots is supported by the recovery of *M. macroura*, which is described as a vital element in their habitat as a source of food and cover (*Velazquez & Heil, 1996*; *Hunter & Cresswell, 2015*; *Rizo-Aguilar et al., 2015*; *Osuna et al., 2022*). In the five burned plots where the number of latrines increased over the sampling year, the height of *M. macroura* exceeded 0.9 m. The reported heights of 1 m, excluding the burned plot of site 9, indicated an exponential increase in RAI (plots at sites 8 and 10). From a landscape perspective, the response of burned plots 8 and 10 can be interpreted by remembering those wildfires are modifying agents that affect the composition and configuration of the area (*Chia et al., 2016*), and that species react according to the availability of resources (*Sutherland & Dickman, 1999*; *Catling, Coops & Burt, 2001*). In this sense, if wildfires occur in areas where the availability and size of patches of habitat suitable for volcano rabbits is already limited, the possibility of finding another patch for refuge is low. Furthermore, there is a high probability that the bunchgrass where they could find refuge was already occupied. Thus, upon the recovery of bunchgrasses in burned areas, volcano rabbits swiftly return to occupy these newly regenerated patches. This is because the emergence of the first shoots of bunchgrass is accompanied by the appearance of fresh pellets of the volcano rabbits in the area. Sites 8 and 10 are located in the forest of Coajomulco, an area with only a few small patches of suitable habitat (*Uriostegui-Velarde et al., 2018*). Given this, because the availability of bunchgrass patches is limited in the forest of Coajomulco, as soon as the burned bunchgrasses recovered, the volcano rabbits returned because there were no other patches with suitable habitat in that landscape. In a specific instance, Site 3 exhibited the highest relative abundance (RAI) of volcano rabbits in unburned areas throughout the year. Remarkably, despite its proximity of only 50 m to the burned plot, no individuals were detected as present. Our interpretation of this event is based on the maximum height of

*M. macroura*, which did not surpass 0.66 m. Additionally, during our monthly visits, cattle were commonly found in this plot. Herding, along with the frequency of fires, impacts the resilience of the vegetation structure (*Díaz-Delgado, 2003*; *Augustine & Derner, 2015*), as reflected by the slow growth of bunchgrasses in this plot.

The analysis of the landscape indicates that the increase in the RAI of the volcano rabbit resulted from greater bunchgrass cover. Thus, greater forest cover and a larger wildfire lead to a decrease in the RAI of the volcano rabbit. Furthermore, a bias analysis of the untreated data returned extreme values in different plots. For example, 33 hectares wildfire occurred in the burned plot at site 3, where latrines were not reported, so this event was a negative factor. Moreover, in nine of the burned plots evaluated, the wildfires affected less than 3.3 hectares and the response of the volcano rabbit was heterogeneous. The interpretation of these results was based on an explanatory analysis, given that the principal component most positively related to the average RAI of the volcano rabbit explained 18% of the variation.

## CONCLUSIONS

We conclude that volcano rabbits can reoccupy burned bunchgrass areas only when other bunchgrasses in the vicinity are occupied regularly or have been used as a refuge during a disturbance. Moreover, we found that volcano rabbits that seek refuge in areas of unburned bunchgrass will occupy the burned areas once *M. macroura* reaches a higher height of 0.9 m and will do so in less than a year. Since the analysis of the landscape and size of the wildfire had little statistical support, we recommend that our results be taken as descriptive, rather than definitive, in nature. The lack of data in annual wildfire reports limited our analysis of the frequency of this phenomenon, which is a cause of extinction of local populations of the volcano rabbit (*Portales et al., 1997*).

Our results indicate that the effects of wildfires on the bunchgrasses of the Ajusco-Chichinautzin Mountain Range should not be considered negative in the context of conservation strategies for the volcano rabbit and its habitat. When this species finds refuges, it can reoccupy burned bunchgrass areas as long as tall *M. macroura* bunchgrass is present. However, human intervention associated with changes in land use (*i.e.*, growing crops and livestock herding) is a crucial element of this interaction. An increase in the frequency and intensity of wildfires can slow the growth of the bunchgrass in the Ajusco-Chichinautzin Mountain Range, which, according to the models presented, would affect the possible reoccupation of the volcano rabbit, moreover with extensive wildfires reduce the possibility that unburned bunchgrass can act as a refuge. Therefore, it is essential to control the frequency and extent of fires in the study area to avoid affecting the resilience of the ecosystem.

Despite discovering evidence of the positive effects of bunchgrass on its reoccupation by the volcano rabbit, we do not recommend prescribed fires to manage the volcano rabbit's habitat. Instead, to obtain more accurate information about the responses of volcano rabbit populations to wildfires, we recommend carrying out more extensive monitoring to analyze the influence of this phenomenon on the composition and configuration of heterogeneous landscapes.

## ACKNOWLEDGEMENTS

We thank M. Bonilla Moheno, V. Farías González, S. Gallina Tessaro, R. Reyna Hurtado and E. Pineda Arredondo for offering useful comments on the manuscript. We are grateful to the laboratory group at the Facultad de Ciencias Biológicas of the Universidad Autónoma del Estado de Morelos for helping with the field work. The Comisión Nacional Forestal (CONAFOR) and Comisión de Recursos Naturales Regional 2 (CORENA R2) generously provided information on wildfires. We also appreciate the comments of the two anonymous reviewers who helped to improve the clarity of the manuscript.

### Funding

The authors received no funding for this work.

### Competing Interests

This research is part of the Doctoral dissertation of Juan Manuel Uriostegui Velarde (CONACyT scholarship 281981) presented at the Instituto de Ecología, A. C.

### Author Contributions

- Juan M. Uriostegui-Velarde conceived and designed the experiments, performed the experiments, analyzed the data, prepared figures and/or tables, authored or reviewed drafts of the article, and approved the final draft.
- Alberto González-Romero conceived and designed the experiments, authored or reviewed drafts of the article, and approved the final draft.
- Areli Rizo-Aguilar conceived and designed the experiments, authored or reviewed drafts of the article, and approved the final draft.
- Dennia Brito-González conceived and designed the experiments, performed the experiments, authored or reviewed drafts of the article, and approved the final draft.
- José Antonio Guerrero conceived and designed the experiments, analyzed the data, authored or reviewed drafts of the article, and approved the final draft.

### Field Study Permissions

The following information was supplied relating to field study approvals (*i.e.*, approving body and any reference numbers):

Field trips were made with the accompaniment of local people. A special permit was not necessary because organisms were not handled, no derivatives of the species were collected, the areas where the samples were taken are located outside the protected natural areas and a group of data was obtained from a local monitoring program executed by a co-author.

### Data Availability

The data are available in the Supplemental Files.

## Supplemental Information

Supplemental information for this article can be found online at http://dx.doi.org/10.7717/peerj.17510#supplemental-information.

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
