# Peer review of "Response of the subalpine bunchgrasses to wildfires and its effects in the relative abundance of the volcano rabbit in the Ajusco-Chichinautzin Mountain Range"

_PeerJ, doi:10.7717/peerj.17510_

## Round 0.1 · original submission · Major Revisions

The manuscript was reviewed by two independent reviewers and both agree that the topic is relevant. Reviewer#1 made comments about the way sentences were organized (especially in the introduction). The same reviewer made suggestions to improve tables and figures. Reviewer#2 provided suggestions for the methods section as well as for figures. Both reviewers provided detailed comments that will improve the manuscript. I hope you find reviewers comments useful.

Reviewer 1 ·

Basic reporting

The language and grammar of this paper is reasonably good, however many sentences (especially in the introduction) are vague and content is a bit disorganized within and among sentences. This makes it difficult to understand the true meaning and how concepts are tied together in support of the research question. The overall approach to setting up the research question is good, but I recommend going back through each sentence and make sure it is as clear and precise as possible. See specific comments by line number below.

1. Throughout – be careful with terminology – the data collected were counts of latrines, which would be an index of abundance, not abundance itself. Although animal sign is commonly used as a de facto index, the authors did not provide evidence that latrines are an actual validated index of abundance for volcano rabbits, so care is needed when using terms such as abundance and recolonization.
• Line 53: The effects of what? On animals, plants? Ecosystems?
• Line 54-55 – Something seems to be missing in this sentence, but lines 56-57 include too many divergent topics. Are the individuals that cannot escape wildfire plants?, but the second part of the sentence jumps to animals modifying their diets. This whole paragraph could be reorganized to first discuss wildfire effects on plants and ecosystems, then on animals.
• Line 59-60 – These 2 sentences need to be linked – what is the adaptation? This section jumps from talking about animal diets to ecosystems. Again, organizing the introduction by ecological scale, type of organisms. Also, benefits and detriments of fire seem to be jumbled in these paragraphs. It would be easier on the reader to first talk about one and then the other.
• Line 63: Are “wildfires” per se actually a management strategy? Do you mean letting them burn? Or prescribed burns?
• Line 67: Give examples of the negative effects on vegetation structure.
• Line 69: Remove “in” before “fewer”
• Lines 71-72 – It is unclear whether you are saying that is easier or harder for small mammals to find refuges with wildfire. Also, provide an explanation,
• Line 75: Provide the scientific name for organism at first mention (in this case snowshoe hares). Also, “hares” is the plural form.
• Line 77: The statement “Small mammal richness increases when the forest is mature, 76 years after a wildfire” is one example finding, so it would be better to say “For example, in [insert location], small mammal richness increased when the forest matured at 76 years (reference).
• Line 79: Explain – the effect of wildfires on what?
• Line 80-81: Use as precise language as possible. By “survival of the species” do you actually mean survival or individuals, presence or sustainability of the population, or whether a species becomes extinct (hence the species does not “survive”)? Which of these did Banks et al. (2011) actually study? The language was unclear.
• Line 83-84 –What specifically do you mean by “resilience of vegetation”? It grows back after fire, it isn’t affected by fire, etc.? How does resilience boost recolonization? Link the concepts together more directly.
• Line 87: provide the full scientific name of organisms the first time they are mentioned in text
• Lines 93-94: This sentence needs rewriting because too many concepts are loosely linked together. Again, by “survival of species”, are you talking about vulnerability to extinction of an entire species? Survival of a species is imprecise.
• Lines 97-99 – Why was a reference to a conference presentation cited? Should this be a personal communication? If so, provide the individual’s name and affiliation.
• Line 105 – use past tense
• Line 107-108 – Rephrase, and provide a stronger rationale than “this is of interest”.
• • Lines 111-115 – In this paragraph, provide very clear hypotheses/predictions that can actually be tested with the data you collected and that can be shown statistically. Provide a hypothesis or prediction for each variable – how did you expect the number of latrines response to burned vs. unburned plots, distance between plots, grass height, time since burn, habitat and any interactions. Recolonization itself was never measured (or if you believe it was, need to define the metric for colonization and how it was measured and analyzed), so predictions can only be about what you measured. In the discussion you can relate your findings to recolonization.
o Figures 3 – 5 Results are clearer and reflect statistical analyses better if they include only 1 panel with one point per time period for burned and one for unburned with a standard error bar. Individual experimental units are provide only descriptive data that cannot be objectively interpreted.
o For figures, make sure the axes labels are large enough to see.
o Table 2-3 – The raw statistical tables are not needed. Instead, provide the test-statistic and p-value in the text when summarizing the findings of the tests. Number in tables have too many significant digits/decimal points

Experimental design

This paper compares an index of abundance (# of latrine sites) of the volcano rabbit, which is threatened with extinction, between 10 burned and unburned patches monthly over a period of a year post-fire, and relates the index to grass height and habitat. The sampling design was generally simple and straight forward, and the analyses provided basic information that showed latrine abundance increased with time since fire on burned sites and was higher with taller grass. Because volcano rabbits are understudied and fire is a potential threat to their limited habitat, this study provides useful data for conservation efforts. However, some aspects of the design and methods were a little unclear, and aspects of the statistical analyses could be improved.

• Lines 127-130 – Provide more information about the study area. Several different climates were described, but not how they were distributed across the study area relative to sites and plots. What is the topography and soils like? Were the sample plots positioned in different climates?
• Lines 137-138: How sites and plots were selected wasn't completely clear. To be selected, did both burned and unburned have to have old volcano rabbit sign, and surveys only counted new sign? Could this bias site selection in any way? From the results and discussion, it sounds like some sites did not have any new latrines post fire, but what about unburned sites? How were new latrines and old rabbit “sign” that was used to select the sites differentiated during the surveys?
• Lines 149-151 – I didn’t understand this sentence. Also provide support that 25 m plots 50 m apart are independent based on home range size or movements or previous research on volcano rabbits or similar species.
• Lines 146 –Do latrines of volcano rabbits occur in certain types of areas (resting, feeding, more cover, etc.) within their home range or are they random? Need to give rationale and links between latrines and abundance. Make sure to be clear that latrines are not abundance, but an index of abundance.
• Line 172: It is helpful to the reader when describing statistical analyses to explicitly explain what the dependent variable is (# of latrines per 25 x 25 m plot) and what the independent variables are (burned vs. unburned, distance between plots as fixed effects and time since burn as a random effect). Explain the variable “distance between plots” and how it was expected to influence latrine abundance. What about interactions? Why wasn’t grass height in this model and instead included in a separate model? This would increase type I error rate.
• Paragraph Line 184 - Why weren’t the landscape variables included in the same model as grass height and burned vs. unburned? Also, explain the independent variables relative to habitat more clearly

Validity of the findings

The results are useful, but they were quite wordy and difficult to read. I suggest combining all independent variables into one model and using simple, clear statements about how latrine abundance responded to each independent variable.

• Line 208 –To better compare with other studies, convert # latrines per plot to #/m2.
• Line 213 – This is a very complex way to say simply that “Unburned plots had more volcano rabbit latrines that unburned plots (F = , P = ), but distance between plots did not influence latrine density (F and p) ” State the statistical conclusions as clearly and succinctly as possible throughout this paragraph and results in general.
• Line 324: Provide only data averages or slopes – individual plot data are not objectively interpretable.
• Line 324: Reduce decimal places of t and p. Work to state statistical result more clearly and simply, such as “As bunchgrass height increased, so did the density of volcano rabbit latrines”. Better to combine all independent variables into one model and use model selection procedures to determine which are most important. Interactions might also be important but weren’t explored.
• Line 282 – Be careful with broad sweeping statements based on a small study that just measured relative abundance of latrines. Any reference to recolonization, etc., should be posed as a possible interpretation, not as a specific finding, unless recolonization was actually measured and analyzed
• Line 287, 289 – This sentence contains too many unrelated items in a list that makes is confusing.
• Line 293-294 – Make sure this is just a possible interpretation from the data, because latrine abundance was all that was actually measured.
• Line 306, 323, 336 – Best to focus on means and trends and not individual plots.
• 329-330 – The results show an increase in latrines over time in burned areas, but how is "immediately" quantified?

Additional comments

no comment

Reviewer 2 ·

Basic reporting

The manuscript “Response of the subalpine bunchgrasses to wildfires and its effects in the relative abundance of the volcano rabbit in the Ajusco-Chichinautzin Mountain Range” (#76080), studied a threatened species (Romerolagus diazi), a species habitat specialist on bunchgrass communities. The authors analysed a comparative between burnt and non-burnt plots. Although, the statistical results are not conclusive, their findings indicate the direction for future projects, considering extensive monitoring to obtain a greater number of samples that contributes to consolidating the models presented.

The introduction is good. The references is very good and the history very well explained although I think that wildfire is always negative effects and the ecosystems, at least on oceanic islands, it is not adapted to fire and produce dramatic effects on all endemic species

Experimental design

With respect to the method. I have some doubts about the field study (see some annotations below). I think the method needs to be better written, specifically with density of rabbit. I think that it is important know rabbit density for tested the magnitude of the problem. There are different method for tested density of rabbit.

Validity of the findings

With respect to the results, the authors could also testing the significant differences between burned and unburned plots using a simple parametric or nonparametric statistics (t-Student or Wilcoxon test, Table 1).
I don’t understand why in the line 260, the authors said that average latrines in burnt plots was 0 to 27.7 but before (lines 210 - 211), the authors said that in burnt plots the average was 99.5 (SE = 38.1) latrines, with a minimum value of 0 and maximum of 332 latrines.

With respect the Figures, it well understood and are well developed although, I don´t understand good the figure 2 , about land use and vegetation types present within a radius of 500 m around the burnt plots. Burnt plots are represented in red and unburnt plots are in black. But it seems that the vegetation type does not coincide in plots of both treatments (S2, S7, S10, …). Maybe by scale….

The discussion is well written and easy to read. For me it is the best of the work together with the introduction.

Additional comments

Some annotations:

Line 36: Include the genus species (M. macroura). This is the first time it has been named.
Lines 53-61: I think that wildfire is always negative effects and the ecosystems, at least on oceanic islands, it is not adapted to fire and produce dramatic effects.
Line 143: Why 10 sites? I think that it is few data for do good statistical analyses.
Lines 157-160: Didn't the colour of the latrines disappear if it rained?
Line 161: There are different methods for estimated rabbit density. Please, see Cooke et al. 2008; Mutze et al. 2014; Rouco et al. 2016; Cubas et al. 2018; 2019)
Counting latrines is not a very efficient method since it does not estimate the number of individuals. For example, in European rabbits (Oryctolagus cuniculus) one latrine belongs to one family.
Lines 161-171: The authors could also make a comparative analysis between burned and unburned plots (t-Student / Wilcoxon).
Lines 175-176: …”Only averaged in the burned sites as a proxy of the recovery of the bunchgrasses”… It is not clear to me what the effect of the plants on the burned plots is like. Did they disappear completely after the fire? Or did the main trunks remain?
Although the authors analyse the effect of fire on volcano rabbit, I believe that the marked plots should be analysed in parallel with the vegetation cover, rock, bare soil,... Have you data about this (vegetation cover, rock cover, slope, bare soil,…)? This variables could be relationship it density of latrines.
Lines 208-212: The LME model revealed an effect in the latrines, but, are significant different the number of latrines between treatment (t-Student or Wilcoxon)?
Line 260: The authors said that average latrines in burnt plots was 0 to 27.7… Please check with respect to lines before (210 -211).

---

## Round 0.2 · Minor Revisions

The reviewers are unavailable but I have checked the reviewers’ suggestions, the authors’ response letter, and the revised manuscript and I found that the manuscript still needs further improvements. Below I provide some comments:

As Reviewer#1 commented, the language and grammar of the manuscript needed revision. I noticed that the authors improved the clarity of the text in some parts, but I believe that overall, the text still needs revision/corrections. Can authors revise the text for grammar and language corrections?

Some examples of sentences that still need correction:
1) Introduction (lines 59-60): Some ecosystems have can tolerate wildfires.

2) Introduction (lines 71-74): The fact that an ecosystem can burn frequently also leads to negative effects on the structure and composition of the vegetation, because it interrupts the vegetation successional cycle, reduces plant diversity, and results in vegetation association homogeneous.

3) Study area (line 195): “Seal level”?

4) Study area (lines 196-197): “the soil type is andosol in lithosol”? This reads very confusingly. Please revise the sentence.

5) Relative Abundance Index of the Volcano Rabbit: “Volcano rabbits form colonies and do not choose from their burrows,..”. This sentence does not make sense, please revise it.

6) Figure 2: Legend needs language/grammar revision. Does “Lineal model” exist? Is it not linear model?

Authors explained in the response letter why they did not include landscape variables in the same model as grass height and burned vs. unburned. However, their response must also be included in the main text.

Figure 3: I also found that the authors did not address the suggestions on Figure 3. Reviewer#1 requested “1 panel with one point per time period for burned and one for unburned with a standard error bar”. Also, the figure legend needs to explain what the shaded area means.

**Language Note:** The Academic Editor has identified that the English language must be improved. PeerJ can provide language editing services - please contact us at [email protected] for pricing (be sure to provide your manuscript number and title). Alternatively, you should make your own arrangements to improve the language quality and provide details in your response letter. – PeerJ Staff

---

## Round 0.3 · accepted · Accept

I have checked the revised manuscript and confirm that the authors have implemented all requested changes.